# Factors associated with adverse childhood experiences in Scottish children: a prospective cohort study

Louise Marryat, John Frank

► Additional material is published online only. To view please visit the journal online (http://dx.doi.org/10.1136/bmjpo-2018-000340).

Farr Institute/Scottish Collaboration for Public Health Research and Policy, University of Edinburgh, Edinburgh, UK

**Correspondence to**
Dr Louise Marryat; louise.marryat@ed.ac.uk

## ABSTRACT

**Background and objectives** Adverse childhood experiences (ACEs) have been associated with a range of poorer health and social outcomes throughout the life course; however, to date they have primarily been conducted retrospectively in adulthood. This paper sets out to determine the prevalence of ACEs at age 8 in a recent prospective birth cohort and examine associations between risk factors in the first year and cumulative ACEs.

**Design** This study uses the Growing Up in Scotland Birth Cohort 1, in which children born in Scotland in 2004/5 were identified using Child Benefit Records and followed up for 7 years (n.3119). ACE scores and sample characteristics were calculated and described. Logistic regression models were fitted to explore associations between risk factors (sex, mother's age and education, household income, area level deprivation and urban/rural indicator) and ACE scores.

**Results** Seven ACEs (or proxies) were assessed: physical abuse, domestic violence, substance abuse, mental illness, parental separation, parental incarceration and emotional neglect. Instances of sexual abuse were too few to be reported. Emotional abuse and physical neglect could not be gathered. Around two-thirds of children had experienced one or more ACE, with 10% experiencing three or more in their lifetime. Higher ACE scores were associated with being male, having a young mother, low income and urban areas.

**Conclusions** Using prospective data, the majority of children born in 2004/2005 in Scotland experienced at least one ACE by age 8, although three ACEs could not be assessed in this cohort. ACEs were highly correlated with socioeconomic disadvantage in the first year of life.

## INTRODUCTION

The adverse childhood experiences (ACEs) scale was first explored with US adults, who were asked a series of questions covering childhood psychological, physical and sexual abuse and household dysfunction. Around half (52.1%) reported experiencing at least one item.[1] Evidence from England and Wales, respectively, showed similar results.[2,3] A study from New Zealand exploring ACEs in the 1970s suggested that c.58% in a prospective study and c.65% in a retrospective study experienced at least one ACE.[4]

### What is already known on this topic?

► Adverse childhood experiences (ACEs) have been found to be commonly reported across adult populations. Limited evidence suggests that higher levels of ACEs are found among adults who had a younger mother and who live in more deprived neighbourhoods.
► Lower levels have been seen in older populations, white or Asian populations and among graduates.

### What this study hopes to add?

► ACEs were associated with being male, low income, younger mothers and urban areas in a current child cohort.

Living in adverse socioeconomic circumstances during childhood has a demonstrated association with later physical and mental health outcomes.[5–7] ACEs go beyond this to look at other adversities, for example, abuse and parental imprisonment. While there is likely to be a substantial overlap with deprivation, this is generally unknown, although evidence of associations with individual measures does exist.[5,8,9] The original ACE study included only adults with private health insurance, suggesting that, as adults, this group were relatively affluent.[1] Neighbourhood deprivation has been associated with increased levels of ACEs;[10] however, Bellis *et al* only found deprivation to be associated with having four or more ACEs.[11] Higher levels of ACEs have been associated with having a younger mother,[11] while lower levels have been found among older people, white or Asian people and graduates.[1]

ACEs have been linked to adverse outcomes in childhood and adulthood:[1,2,12,13] they have been associated with poorer self-rated health, premature mortality, suicide attempts, depression, ischaemic heart disease, cancer, chronic lung disease, skeletal fractures, liver disease, fetal death and chronic obstructive pulmonary

disease.[1 13–17] ACEs have also been associated with risky lifestyle behaviours, for example, drug and alcohol use, smoking and high numbers of sexual partners[1] as well as job, financial and reproductive problems.[15] They have been linked to multimorbidity in adulthood.[1]

Despite the current interest in ACEs and their seeming importance in relation to subsequent outcomes, we know little about modern ACEs. The capturing of ACEs in the general population is difficult and relies on retrospective data collected in adulthood. The aims of this paper are to explore to what extent ACEs could be determined using prospective cohort data and what prevalence levels look like in a recent population. This approach allows exploration of the predictors of ACEs, normally not available in retrospective data.

## METHOD

Child and parent interview data were taken from the first seven waves of the Growing Up in Scotland (GUS) study, covering the first 8 years of the child's life. This study, funded by the Scottish Government, tracks the lives of children from birth through to their teenage years and beyond and collects a wide range of information, including cognitive, social, emotional and behavioural development, health and well-being, childcare, education and parenting and social networks. The full design has been reported elsewhere.[18] The sample was taken from child benefit records, which at the time of sampling included 97% of the Scottish population with children. Data zones (the key small-area geographic statistic in Scotland, each containing 500–1000 people and nested within Local Authorities) were aggregated until each area had an average of 57 live births per year, based on the previous 3 years of data, which was estimated to provide the required sample size. These primary sampling units were then stratified by Local Authority (n.32) and then by Scottish Index of Multiple Deprivation Score (a measure of relative area level deprivation), before 130 points were chosen at random. Prior to final sampling, the Department of Work and Pensions removed any 'sensitive cases' (eg, where there had been a death in the immediate family) and any cases which had been sampled in the previous 3 years (c.5%). They then sampled all babies within the selected points which met the date of birth criteria. In cases where more than one child met the criteria, one child was selected at random. Where children were found not to be living with their natural parent, they were followed up where possible in foster care or kinship care. Sweep 1 took place in 2005/2006 when the children were 10 months old with 5217 children, comprising 81% of the eligible children.[19] At Sweep 7, 3456 children remained (66% of sweep 1 children), with a target interview date of 94.5 months old.[18] Table 1 details the missing cases. Cases were only included were the participant had the relevant ACE and demographic data, of which there were 3119 (90.2% of all Sweep 7 subjects). Exploratory analyses indicated that

**Table 1** Proportion of children who were included or missing at sweep 7 by demographic characteristics

| | Missing from sample (%) | Included in sample (%) |
|---|---|---|
| **Family type** | | |
| Lone parent | 29.7 | 13.2 |
| Living with partner | 70.3 | 86.8 |
| **Age of mother at birth of cohort child** | | |
| Under 20 | 12.6 | 3.8 |
| 20–29 | 49.5 | 35.6 |
| 30–39 | 35.3 | 56.6 |
| 40 or older | 2.6 | 4.0 |
| **Income quintile** | | |
| Bottom quintile | 31.0 | 14.5 |
| 2 | 22.7 | 19.3 |
| 3 | 16.0 | 19.5 |
| 4 | 16.2 | 24.4 |
| Top quintile | 14.1 | 22.3 |
| **Scottish Index of Multiple Deprivation Quintiles** | | |
| 5—Most deprived | 32.1 | 17.5 |
| 4 | 19.4 | 17.1 |
| 3 | 16.7 | 21.7 |
| 2 | 17.7 | 21.6 |
| 1—Least deprived | 14.2 | 22.1 |
| **Education level of mother** | | |
| No qualifications | 14.7 | 6.2 |
| Standard Grades or above | 26.1 | 15.1 |
| Highers or above | 59.2 | 78.7 |
| **Child ethnic group** | | |
| White | 93.5 | 96.5 |
| Non-white | 6.5 | 3.5 |
| **Child sex** | | |
| Male | 53.1 | 50.6 |
| Female | 46.9 | 49.4 |
| **Urban/rural indicator** | | |
| Large urban | 40.1 | 36.7 |
| Other urban | 34.6 | 29.4 |
| Small accessible towns | 9.2 | 9.8 |
| Small remote towns | 2.5 | 3.2 |
| Accessible rural | 10.5 | 15.4 |
| Remote rural | 3.1 | 5.4 |
| *Base* | *1761* | *3456* |

respondents missing data at this stage were more likely to be younger mothers, those living in more deprived areas and those in urban areas. Cases without full data were not used: no imputation was used. Longitudinal weights, produced by the survey team at ScotCen, were

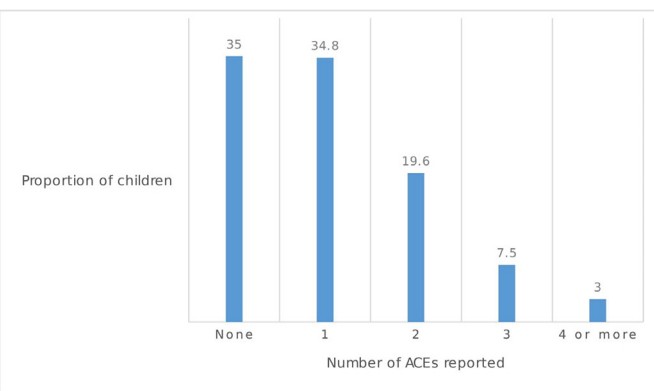

**Figure 1** Number of ACEs reported at age 8. ACEs, adverse childhood experiences.

used, and the stratification of the sample was accounted for using the Complex Samples module within SPSS24.[20] Full information on weighting can be found in the data user guide.[18]

Data from GUS were mapped onto the ACE questions (online supplementary table 1) and sociodemographic information from the first sweep were combined into one dataset. Since ACE-related questionnaire items were asked inconsistently throughout the seven sweeps of the study, an 8-year cumulative incidence of ACEs was derived by summing the ACEs present for each child throughout the study period. Seven ACEs (or proxies for them) were available: physical abuse, domestic violence, substance abuse, mental illness, parental separation, parental incarceration and emotional neglect. Information on Emotional abuse and Neglect was not available. Sexual abuse was reported in response to an open-ended question; however, personal communications with the survey team revealed that fewer than five participants had responded positively to this, and it was thus excluded.

### Ethics
Each sweep of data collection was subject to medical ethical review by the Scotland 'A' MREC committee. All parents interviewed (and Children, for the child interview) gave their informed consent prior to inclusion in the study. The current analyses were subject to ethical review in the University of Edinburgh.

### Analyses
Binary variables of one or more ACE (vs none) (1+ ACE) and three or more ACEs (vs <3) (3+ ACEs) were derived. Three or more ACEs were explored, rather than four or more, due to the number of ACEs assessed being fewer and children having had less time to accumulate them. Sociodemographic data were described and Spearman correlations between them were explored. All correlations were of a weak to medium strength (online supplementary table 2).

Multiple univariable logistic regression models were fitted predicting the odds of having 1+ and 3+ ACEs, respectively (table 3). Predictors were taken from the current literature: sex, ethnicity, income, age of mother

at the birth of the first child and mother's educational qualifications. Where univariable associations reached the criterion of p<0.1, these variables were put into the multivariable model.

### Patient and public involvement statement
This paper uses secondary data analysis of the first seven sweeps of the GUS study. The original study was designed in conjunction with the Scottish Government, academics and third sector organisations. The current analyses have had input from the Scottish Government ACEs team, have been presented and discussed at an NHS event as well as being discussed with a variety of third sector organisations representing and working with people who have experienced ACEs. Further dissemination of the work is planned to practitioners working with people who have experienced ACEs as well as to the general public through social media and national media.

## RESULTS
Around two thirds (65%) of children experienced 1+ ACE, with 10% experiencing 3+ at age 8 (figure 1).

The most common ACEs were parental mental health problems (35.4%) and having parents who were separated or divorced (32.1%). The proportion reporting frequent physical punishment was high (22.2%). Few children experienced parental imprisonment (0.4%) (table 2).

ACEs varied by household income quintiles: 1% of children in households in the top income quintile had 4+ ACEs, compared with 10.8% in the lowest income group (table 3).

Boys were more likely to experience 3+ ACEs, as were those with younger mothers, mothers with fewer educational qualifications and children living in more deprived areas. A higher proportion of children from a white UK

| **Table 2** Proportion of children within each ACE category | |
|---|---|
| **ACE field** | **Weighted percentage** |
| ACE1: emotional abuse | – |
| ACE2: corporal punishment | 22.2% |
| ACE3: sexual abuse | – |
| ACE4: not loved or supported | 20% |
| ACE5: neglect | – |
| ACE6: parents ever divorced or separated | 32.1% |
| ACE7: domestic violence | 9.1% |
| ACE8: drug or alcohol misuse | 14.0% |
| ACE9: mental health problems | 35.4% |
| ACE10: parent in prison | 0.4% |
| Any ACE reported | 65.0% |
| *Base* | *3119* |

ACE, adverse childhood experience.

**Table 3** Proportion of children in each ACE category by demographics

| | Sample size | No. of ACEs (%) | | | | | |
| --- | --- | --- | --- | --- | --- | --- | --- |
| | | None | 1 | 2 | 3 | 4 or more | χ² . p value |
| **Sex** | | | | | | | |
| Male | 1609 | 32.9 | 34.6 | 20.4 | 8.4 | 3.8 | 0.004 |
| Female | 1510 | 37.3 | 35.1 | 18.8 | 6.5 | 2.2 | |
| **Ethnicity** | | | | | | | |
| White | 2982 | 34.7 | 34.7 | 19.9 | 7.6 | 3.1 | 0.07 |
| Non-white | 135 | 43.6 | 36.7 | 13.4 | 5.6 | 0.6 | |
| **Education of mother** | | | | | | | |
| Degree or higher | 842 | 51.4 | 32.9 | 11.8 | 2.8 | 1.1 | <0.001 |
| Vocational qualification | 1163 | 35 | 35.9 | 19.2 | 7.9 | 2 | |
| Highers | 240 | 36.4 | 32.2 | 24.9 | 5.9 | 0.6 | |
| Standard grade | 555 | 18.7 | 38.2 | 23.3 | 12.3 | 7.4 | |
| No qualifications | 291 | 15.6 | 32.5 | 33.3 | 11.9 | 6.7 | |
| Other | 24 | 56.3 | 30.7 | 8 | 5 | 0 | |
| **Age of mother at birth of 1st child** | | | | | | | |
| Under 20 | 541 | 12.5 | 30.1 | 31.5 | 16.3 | 9.5 | <0.001 |
| 20 to 29 | 1590 | 34.8 | 35.9 | 20.1 | 7.0 | 2.3 | |
| 30 to 39 | 945 | 48.3 | 36.2 | 11.4 | 3.5 | 0.6 | |
| 40 or over | 35 | 46.4 | 28.8 | 20.2 | 2.4 | 2.2 | |
| **Household equivalised income quintile** | | | | | | | |
| Bottom (£<8410) | 623 | 8 | 27 | 34.5 | 19.7 | 10.8 | <0.001 |
| 2nd quintile (£8410–£13 749) | 587 | 26.6 | 33.6 | 26.1 | 10.6 | 3.2 | |
| 3rd quintile (£13 750-£21 784) | 518 | 33.8 | 38.6 | 18.3 | 8.3 | 1 | |
| 4th quintile (£21 785–£33 570) | 592 | 42.2 | 36 | 17.1 | 4.2 | 0.5 | |
| Top quintile (£≥£33 571) | 513 | 52.8 | 32.7 | 11.1 | 2.3 | 1 | |
| **Scottish Index of Multiple Deprivation area quintile** | | | | | | | |
| 1—Most deprived | 745 | 17.2 | 37.6 | 29.1 | 10.9 | 5.3 | <0.001 |
| 2 | 583 | 30.3 | 35.9 | 19.2 | 10.9 | 3.7 | |
| 3 | 637 | 34.6 | 35.3 | 22 | 5.6 | 2.5 | |
| 4 | 593 | 43 | 35.2 | 14.6 | 5.3 | 2.0 | |
| 5—Least deprived | 561 | 55.7 | 29.3 | 10.2 | 3.8 | 1.0 | |
| **Urban/rural classification** | | | | | | | |
| Urban | 2567 | 32.6 | 34.9 | 20.8 | 8.3 | 3.4 | <0.001 |
| Rural | 552 | 46.3 | 34.5 | 14.3 | 3.6 | 1.3 | |

ACE, adverse childhood experience.

background, compared with other ethnic groups, experienced 3+ ACEs, although the low proportion (4.3%) of children from non-white-UK backgrounds limited the power of this comparison. Children in urban areas were more likely to have experienced a higher number of ACEs (3.4% vs 1.3% in rural areas).

Independent associations were explored between risk factors in the first year of life and having 1+ or 3+ ACEs, respectively. The multivariable model indicated that risk factors predicting having 1+ ACE at age 8 were living in a lower income group, in particular living in the lowest income quintile (OR 7.11); being male; having a mother with lower educational qualifications; having a mother who was under 20 or over 40 at the birth of her first child; living in an area with higher levels of deprivation or in an urban area (table 4).

The model predicting having 3+ ACEs at age eight was very similar. In this multivariable model, maternal educational qualifications did not meet our criteria for statistical significance, although it is worth noting that the ORs were similar to the previous model. Once again, the strongest predictor of experiencing 3+ ACEs was living in a household in the lowest income quintile, where odds were 5.7 times higher (table 4).

**Table 4** Univariable and multivariable logistic regression models predicting having 1+ and 3+ ACEs, respectively, at age 8

| | Univariable: 1+ ACE | | Multivariable 1+ ACE | | Univariable: 3+ ACE | | Multivariable 3+ ACE | |
|---|---|---|---|---|---|---|---|---|
| | OR | 95% CI | OR | 95% CI | OR | 95% CI | OR | 95% CI |
| **Sex of child** | | | | | | | | |
| Male | 1.21 | 1.03 to 1.44 | 1.31 | 1.09 to 1.56 | 1.45 | 1.11 to 1.89 | 1.51 | 1.15 to 1.98 |
| Female | – | – | – | – | – | – | – | – |
| **Education of mother** | | | | | | | | |
| Degree or higher | 0.18 | 0.11 to 0.27 | 0.87 | 0.54 to 1.41 | 0.18 | 0.10 to 0.33 | 0.86 | 0.38 to 1.92 |
| Vocational qualification | 0.34 | 0.22 to 0.53 | 1.07 | 0.67 to 1.73 | 0.48 | 0.30 to 0.79 | 1.10 | 0.58 to 2.10 |
| Highers | 0.32 | 0.20 to 0.53 | 0.94 | 0.55 to 1.63 | 0.30 | 0.13 to 0.67 | 0.43 | 0.16 to 1.16 |
| Standard Grade | 0.80 | 0.51 to 1.27 | 1.72 | 1.07 to 2.78 | 1.08 | 0.67 to 1.73 | 1.56 | 0.87 to 2.78 |
| Other | 0.14 | 0.05 to 0.43 | 0.41 | 0.12 to 1.45 | 0.23 | 0.03 to 1.85 | 0.54 | 0.05 to 6.53 |
| No qualifications | – | – | – | – | – | – | – | – |
| **Age of mother at birth of child** | | | | | | | | |
| Under 20 | 6.07 | 3.16 to 11.66 | 1.44 | 0.69 to 3.01 | 7.26 | 1.58 to 33.37 | 2.19 | 0.45 to 10.57 |
| 20–29 | 1.62 | 0.87 to 3.02 | 0.75 | 0.37 to 1.52 | 2.12 | 0.49 to 9.16 | 1.06 | 0.23 to 4.82 |
| 30–39 | 0.93 | 0.49 to 1.75 | 0.74 | 0.35 to 1.59 | 0.88 | 0.20 to 3.88 | 0.70 | 0.15 to 3.25 |
| 40 or over | – | – | – | – | – | – | – | – |
| **Household equivalised income quintile** | | | | | | | | |
| Bottom (£<8410) | 15.05 | 9.67 to 23.43 | 7.11 | 4.53 to 11.17 | 11.99 | 7.02 to 20.47 | 5.73 | 2.82 to 11.64 |
| 2nd quintile (£8410–£13 749) | 3.16 | 2.36 to 4.22 | 1.96 | 1.52 to 2.53 | 3.90 | 2.12 to 7.16 | 2.45 | 1.15 to 5.20 |
| 3rd quintile (£13 750–£21,784) | 2.07 | 1.59 to 2.69 | 1.52 | 1.17 to 1.97 | 2.97 | 1.70 to 5.20 | 2.26 | 1.17 to 4.35 |
| 4th quintile (£21 785–£33 570) | 1.59 | 1.23 to 2.05 | 1.38 | 1.07 to 1.79 | 1.46 | 0.80 to 2.65 | 1.31 | 0.71 to 2.41 |
| Top quintile (£≥£33 571) | – | – | – | – | – | – | – | – |
| **Scottish Index of Multiple Deprivation area quintile** | | | | | | | | |
| 1—Most deprived | 6.06 | 4.71 to 7.80 | 2.29 | 1.71 to 3.06 | 1.56 | 0.89 to 2.72 | 1.05 | 0.60 to 1.84 |
| 2 | 2.90 | 2.25 to 3.73 | 1.47 | 1.07 to 2.01 | 1.75 | 1.00 to 3.05 | 1.19 | 0.66 to 2.16 |
| 3 | 2.38 | 1.93 to 2.94 | 1.83 | 1.42 to 2.36 | 3.51 | 2.02 to 5.75 | 0.98 | 0.54 to 1.80 |
| 4 | 1.67 | 1.31 to 2.13 | 1.52 | 1.15 to 2.02 | 3.84 | 2.33 to 6.34 | 1.22 | 0.73 to 2.04 |
| 5—Least deprived | – | – | – | – | – | – | – | – |
| **Urban/rural classification** | | | | | | | | |
| Urban | 1.79 | 1.50 to 2.13 | 1.57 | 1.26 to 1.95 | 2.56 | 1.62 to 4.05 | 1.82 | 1.14 to 2.92 |
| Rural | – | – | – | – | – | – | – | – |
| **Ethnicity** | | | | | | | | |
| White | 1.46 | 0.95 to 2.48 | – | – | 1.80 | 0.80 to 4.03 | – | – |
| Non-white | – | – | – | – | – | – | – | – |
| *Sample size* | *2848* | | *2848* | | *2848* | | *2848* | |

ACE, adverse childhood experience.

## DISCUSSION

By 8 years, around two-thirds of children had 1+ ACE, with 1 in 10 children experiencing 3+. Results indicate that children living in more disadvantaged circumstances were more likely to experience ACEs than their more privileged peers.

### Strengths

This paper is unique in using a range of prospective data on ACEs in a current generation of children. GUS covers a representative sample of children from Scotland, providing large enough numbers to explore ACEs by subgroups, combining a mixture of parental and child-reported data, giving a rounded picture of ACEs which have occurred. The inclusion of family background data allows us insight into what early predictors are associated with having higher odds of experiencing ACEs. In addition, families were asked to answer a range of questions about their lives, rather than a specific questionnaire about ACEs: Felitti *et al* found that respondents to the ACE questionnaire were slightly more likely to have reported sexual abuse in a separate questionnaire, suggesting that there may be a bias within the original ACE studies towards people who had experienced ACEs.[1]

### Weaknesses

The original ACE questionnaire has not yet been included in GUS and so some proxy information had to be used, which may not accurately reflect the ACE questionnaire. Two ACEs (emotional abuse and neglect) were not able to be assessed through the GUS questions. In addition, the proportion of parents reporting sexual abuse was so low that we were not able to include this. It is possible that social desirability may lead to some parents not disclosing where ACEs had occurred and, furthermore, if ACEs, such as abuse, happened outside the home, parents may not be aware that they have occurred. All these factors are likely to lead to this study underestimating the prevalence of ACEs. Despite the sample being representative of the population of children in Scotland, some particularly sensitive cases were removed prior to sampling by the Department for Work and Pensions, which likely include children who suffered adversity within the first 8–9 months. GUS suffers from differential attrition, whereby children from the most disadvantaged backgrounds are disproportionately lost to follow-up.[21] Longitudinal weights do make up for this to some extent.

The proportion of children with 1+ ACE was substantially higher than most other retrospective studies, in contrast to findings from New Zealand.[1–4] There are various factors which might explain this difference. Differences in the way questions are phrased and proxies used are likely to affect prevalence reported. Second, there are issues around recall: data in the latter studies were collected up to 40 years after the events took place, compared with continuous collection in GUS. Hardt and Rutter concluded that reports of ACEs in adulthood involve a 'substantial rate of false negatives' due to a lack of very early childhood memory, mood congruent recall biases and the fact that people are only able to recall what they were aware of at the time[22]—for example, some parents keep the incarceration of a parent from their children.[22]

Cohort effects may also play a role: retrospective studies look back at childhoods between the 1950s and 1970s, where parenting and societal norms often differed from the present day: harsh corporal punishment was far more frequent historically and still varies substantially between countries.[8] It is noteworthy that the New Zealand cohort with the most recent data (from childhoods in the 1970s–1980s) displayed the most similar results to the GUS study.[4] Access to data on cohort differences in ACEs within studies is somewhat limited, although one retrospective study, which looked at four cohorts between 1900 and 1978, found evidence that the proportion of children with no ACEs reported increased from 16.7% to 21.6% over that time—although those reporting 4+ ACEs also increased slightly, from 43.1% to 51.1%.[12]

Children in the GUS sample were less likely to have experienced 3+ ACEs, than the USA, New Zealand, Welsh and English studies.[1–4] This may well be due to the difference in age-period examined, where previous studies examined ACEs up to the age of 18, compared with up to the age of eight in GUS, providing less time for children to 'accumulate' ACEs.

Other factors which were associated with having increased odds of having 1+ ACE or 3+ ACEs, respectively, were being male, having a younger mother and living in an urban area. The finding that boys were more likely to experience ACEs is similar to that seen in the previous English retrospective study,[2] but was the opposite of findings from Felitti *et al*, which found that 18% of women experienced 3+ ACEs, compared with 9% of men. This may be due to the two of the items which were unable to be measured (sexual abuse and emotional abuse), both of which are more likely to be experienced by women.[1] The relationship to age of mother was reflected in previous findings: Bellis reported increased odds of higher numbers of ACEs for children born to mothers aged under 20.[11] The exploration of urban/rural differences in ACE counts appears to be a novel contribution to the literature, with no previous evidence produced on these differences, although living in an urban area is independently associated with higher levels of child abuse,[9] drug abuse[23] and heavy alcohol use[24]— all contained within the ACE questions. These models focused on sociodemographic risks for ACEs: future research may wish to explore the explanatory power of factors such as attachment,[25] neurodevelopmental disorders,[26] parenting and parental ACEs.[27 28] Additional research is also needed to find out how many Scottish children experience further ACEs up to age 18 and how this compares to other parts of the world.

## CONCLUSION

Around two-thirds of Scottish 8 year-olds had experienced at least one ACE during their life. While this compares unfavourably to previous ACE studies, measurement differences make it difficult to directly compare—although the ACEs reported in this study are clearly a subset of those reported in most studies. Although a large proportion of children had experienced one ACE, just 10% experienced 3+. ACEs were associated with poverty: children living in the lowest household income quintile had odds around seven times higher of having 1+ ACE than the most affluent children.

**Contributors** LM designed the study, cleaned and analysed the data and drafted the final report. JF advised on study design and analyses. LM and JF both read and agreed the final manuscript and agree to be accountable for all aspects of the work.

**Funding** LM is supported by the Farr Institute at Scotland, which is supported by a 10-funder consortium: Arthritis Research UK, the British Heart Foundation, Cancer Research UK, the Economic and Social Research Council, the Engineering and Physical Sciences Research Council, the Medical Research Council, the National Institute of Health Research, the National Institute for Social Care and Health Research (Welsh Assembly Government), the Chief Scientist Office (Scottish Government Health Directorates) (MRC Grant No. MR/K007017/1). LM sits within, and JF is supported by, the core grant to SCPHRP from the MRC, with half that support from the Scottish Chief Scientist Office (MR/K023209/1).

**Competing interests** None declared.

**Patient consent** Not required.

**Provenance and peer review** Not commissioned; externally peer reviewed.

**Data sharing statement** All data are available through the UK Data Archive.

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
