## [Reviewer comments · BMJ Paediatrics Open]

ARTICLE DETAILS

TITLE (PROVISIONAL)	Factors associated with Adverse Childhood Experiences in Scottish children: a prospective cohort study
AUTHORS	Marryat, Louise; Frank, John

VERSION 1 – REVIEW

REVIEWER	Reviewer name: Sarah J Nevitt Institution and Country: University of Liverpool, United Kingdom Competing interests: I have no competing interests
REVIEW RETURNED	12-Jul-2018

GENERAL COMMENTS	I have conducted a statistical review of the manuscript “What factors are associated with Adverse Childhood Experiences in Scottish children: results from a birth cohort study.” The authors investigate which factors are associated with adverse childhood experiences using a prospective cohort of over 3000 children in Scotland. The authors present the first analysis to investigate ACEs using a prospective approach. I am satisfied that the statistical approach taken by the authors is appropriate for the objectives and data in question. I have made a few comments where I feel wording or terminology is unclear, or where I feel additional information would be helpful for a reader: 1) Introduction, page 4, line 53: “The aims of this paper are to explore to what extent ACEs could be imputed using prospective cohort data,” I’m not sure that ‘imputed’ is quite the right word here, perhaps ‘investigated’ or ‘predicted’ (if the aim is to determine ACEs before they have happened).2) Methods, page 5, line 12 “Data were taken from the first seven waves of the Growing Up in Scotland Study...” Please define the GUS abbreviation here (it is used in the paragraph below).3) Methods, page 5. For context, can the authors please add a sentence or two on the aims/ objectives etc. of the Growing Up in Scotland Study? I assume that the study was conducted for wider objectives than determination of ACEs?4) Methods, page 5, line 14: What is the sample clustered by?5) Methods, page 5, line 54: “Seven ACEs (or proxies for them) were available...” It would help for context to add out of how many possible ACEs. This is implied in the rest of the paragraph but it would be helpful to make it explicitly clear: i.e. “Seven ACEs (or proxies for them) out of 10 possible ACE questions were available...”6) Analyses, page 6, line 21: “Binary variables of 1+ ACE (vs. none) and 3+ACEs (vs. <3) were derived.”
---

Some of the notation here is a little confusing, on first reading I thought this was a calculation (i.e. 1 plus the number of ACE questions). To avoid misunderstanding throughout these methods and results, I suggest using words rather than + signs (i.e. one or more ACE vs no ACEs, at least 3 ACEs vs less than 3 ACEs etc.).

7) Analyses, page 6, line 21: "Data were described and Spearman correlations explored. All correlations were of a weak to medium strength."
Which data is being described here in terms of correlations? I presume that this data was skewed (i.e. non-normally distributed) due to the use of non-parametric Spearman rank correlation?

8) General comment, analysis and results. Please note that logistic regression produces odds ratios (rather than risk ratios) and therefore results should be interpreted in terms of odds rather than risk i.e. please reword 'predicting the risk' (page 6, line 26), "...associated with heightened risk" (page 9, line 3) and so on.

9) Analysis, page 6 and 7: The terms used in the calculation for population attributable risk and the terms in the footnotes of this calculation are different (P and Pe, RR and RRe), please use consistent terminology.

10) Results and Discussion. Minor point, but 65% is not two thirds. Please reword to 'about two thirds' or 'nearly two thirds' or just quote the proportion

11) Results: It would be helpful if the numbers reporting each of the seven ACES could be reported in a table / on a Figure. Most of the results refer to the number of ACEs reported rather than specifics. Some information is given in the second paragraph of the results regarding specific ACEs but complete details of this would be interesting to see

12) Table 2: Please also report numerical results (odds ratios and confidence intervals) for the non-significant results. The magnitude of effect (as well as statistical significance) may be of interest to readers.

13) Table 2: R-squared values (the amount of variability explained by the covariates in the regression model) are relatively low and a lot of variability remains. This is not a shortcoming of the authors or the methods, it is the reality of attempting to find statistical associations where relationships are complex and likely confounded. Do the authors have any thoughts on any factors (known or unknown, perhaps information that was not collected within the GUS) study which could also be influencing ACEs? Could the authors add a couple of sentences to the discussion regarding the remaining variability and, if known, any potential reasons for this?

14) Figure 1 and Figure 2, Table 2 – I assume that 'base' refers to the sample size of the data displayed within the figure/table? Perhaps use a different term for clarity

15) Figure 1: I assume this title should read "Number of ACEs experienced at age eight"?

16) Figure 2: It is quite difficult to read the numbers (I assume proportions %?) on bars, particularly those close together towards the top. Perhaps these % could be displayed next to the bar rather than on top, or in a table below the figure?

17) Results, page 11, line 3 "As poverty appeared to play such an important role in children's likelihood of experiencing ACEs..."

18) Please add details into the background of the association between poverty and ACEs with any references add needed.

19) Results, page 11: Please also provide the numbers of the sample living in relative poverty with and without at least one ACE (so that a reader has enough information to perform the calculation in the methods, if desired).

REVIEWER	Reviewer name: Barry Milne Institution and Country: University of Auckland, New Zealand Competing interests: None
REVIEW RETURNED	30-Jul-2018

GENERAL COMMENTS	This study documents the prevalence of adverse childhood experiences (ACEs) using prospectively-collected data from the Growing Up in Scotland Study, and examines early childhood risk factors for ACE experiences. This is a worthwhile contribution to the literature, but suffers from lack of clarity in some areas. These will need to be addressed before this paper can be considered for publication. Examples of this are:  1. In the final paragraph of the introduction it is stated, “The aims of the paper are to explore to what extent ACEs could be imputed using prospective cohort data, and what prevalence looks like in a recent population”. No imputation occurred from what I could tell. Isn’t the aim just to “determine the prevalence of ACEs using prospective cohort data”? 2. I found the description of the sample confusing. From the description, it seems there were PSUs of aggregated data zones, each with an average of 57 live births/year. How were births chosen from these PSUs, and what role did the stratification by local authority and imd score play? And how many births did this result in sampling, and what steps were there in sampling the 5,217 assessed at sweep 1? What was ‘sensitive’ about the sensitive cases that meant they had to be removed? No doubt the sample has been described many times in Growing Up in Scotland publications, so perhaps a baseline GUS study can be referred to, but the description in this paper still needs to be clearer, in my view. 3. At the end of the first paragraph of the methods it is stated, “Cases without full data were not used: no imputation was used.” The obvious question arises whether the 10% not assessed differ from the 90% assessed. From the discussion and ref 17 it looks like there was differential loss to follow-up. How exactly did the 10% differ from the 90%? 4. It is not stated who reported on the ACEs. In the strengths section of the discussion it is indicated that “a mixture of parental and child-reported data” were used, but this isn’t described anywhere. Who reported on which ACE? 5. In the analyses section it appears that variable selection took place, but Table 2 seems to include all predictors. What was the procedure? My personal preference is for no selection and to show univariable and multivariable associations, but I’ll leave that to the authors. 6. The PAR analysis at the end of the results comes across as a little ad-hoc, and the interpretation assumes the poverty-ACEs association is causal, which is moot. I’d be inclined to remove this unless there’s a specific reason for inclusion that is addressed in the introduction. 7. The discussion mentions longitudinal weights, but the methods do not indicate how these are used (and calculated).
---

	There are also several minor issues that the authors may wish to address:  1. Under 'What the study adds', it is stated that, "At age 8, children had higher levels of one ACE". Higher than what? If 'high' is meant, perhaps state the actual proportion or percentage with 1+ and 4+. 2. The 'GUS' acronym isn't described in the text. Maybe do so in the first line of the methods. 3. I don't understand the 'e' in the PAR formula. 4. The education and deprivation associations in Table 1 aren't discussed in the text, where all other associations are. Why? Also, figure 2 repeats the income findings from Table 1. Why? 5. In Table 2, the r-squared and 'base' (is this the n?) are shown in the odds ratio column. Maybe these could be at the top of the table or footers. 6. Under strengths, it's unclear why the English and Welsh ACE studies are specifically targeted when highlighting the GUS response rate as a strength. Perhaps instead compare the GUS response rate to a range of international studies. 7. In the conclusion, perhaps add "during their life" to the end of the first sentence.
--	---

REVIEWER	Reviewer name: James Doidge Institution and Country: University College London Competing interests: Have authored similar work.
REVIEW RETURNED	15-Aug-2018

GENERAL COMMENTS	This manuscript describes an analysis of risk factors for adverse childhood experiences (ACE) in a Scottish birth cohort, including estimation of the proportion of ACE attributable to poverty. As the authors indicate, there are few prospective studies in the ACE literature. There are also few on the effects of poverty on ACE. Unfortunately, though, the analysis is inadequately reported, poorly integrated with the existing literature, and suffers from several statistical limitations and errors. My major concerns include:  1. The description of the methods is unclear. Initial statements like "derived from child benefit records" (P5, L14) seem to imply that the study relied on administrative data. Then, later in the same paragraph there is a statement "the target interview date was 94.5 months old". Was this for sweep 7? There is no explanation at all of data collection between sweeps 1 and 7. Were participants interviewed at every sweep, or was it some blended design of administrative and interview/survey data? Who was interviewed – the parents and/or children? (it is indicated later that both were interviewed but the details about when and how are not at all clear). 2. Statistical problems include that:  a. Missing data are ignored, despite known/expected associations with ACE. There is no explanation for what happened to 34% of the cohort between sweeps 1 and 7, no comparison of characteristics between those lost and those retained, or those incomplete (another 10%) and those complete.
---

Really, there should be at least an attempt at inverse probability weighting or multiple imputation but even a descriptive analysis is missing.

b. A simple, poorly described, automated variable selection procedure was implemented to build the model according to the significance of individual parameters. Automated variable selection is better suited to building predictive models from large numbers of available predictors, not for adjusting for confounding in causal effect estimation, which appears to be the aim of this analysis (specifically, estimating the effect of poverty on ACE risk). Estimating causal effects requires a conceptual understanding of confounding (i.e. justification for the selection of covariates, more than just that they significantly predict the outcome) and other sources of bias (e.g. missing data), with appropriate statistical methods for addressing them. Crucially, you should not drop confounders or explanatory variables just because they are not significant.

c. Attributable fractions have been calculated using an inappropriate formula; the formula for use when there is no confounding, rather than that for use with adjusted relative risks. See Rockhill B, Newman B, Weinberg C. Use and misuse of the population attributable fraction. *Am J Public Health.* 1998;88(1):15-9; or Gefeller O. Comparison of adjusted attributable risk estimators. *Stat Med.* 1992;11(16):2083-91.

d. The interpretation of results indicates a poor understanding of the differences role of statistic significance testing vs effect sizes, the relationship between odds ratios and relative risks, the interpretation of attributable risk, the interpretation and use of pseudo-R² (mistakenly referred to as R²) in logistic regressions, and the even the implied use of R² when the aim of modelling is not prediction or construction of a full explanatory model.

Given these multiple limitations in the analysis and interpretation of results, I strongly suggest that the authors seek statistical support before revising.

3. The 'mapping' of adverse childhood experiences from the GUS interview responses is poorly described and inconsistent with the literature. For example, the authors seem to have equated smacking "when [the child] has done something wrong" (often or always) with physical abuse. This is highly contentious and not consistent with other definitions used for physical abuse, including the one cited for comparison. The threshold level of >14/units per week that the authors have set for mapping parental alcohol problems, while associated with increased health risks, would not normally be considered indicative of problem drinking in the context of ACEs. Do the authors know what proportion of UK adults/parents drink at this level and how those statistics relate to those recorded in the study? And how these relate to parental alcohol problems recorded in other ACE studies?

4. The prevalences of each ACE are not even reported, making it impossible to assess their consistency with other research or the contribution of each ACE to the number of ACEs.

5. The introduction is entirely missing references to the existing literature on risk factors for ACE and the impact of poverty on ACE. The discussion is dismissively critical of previous research, to the point of seeming unjustified or unkind in places.

Other comments and questions:

6. P2L6 (what this study adds): 'higher' than what? Suggest rephrasing.
7. P3L18 (Design): Logistic regression models were fitted to explore associations – adjusted for what?
8. P4: Opening sentence grammar needs improving at “, and”, e.g. “, who were”. Also, acronyms should not have 's' appended in plural form (but check with journal guidelines).
9. P4, L27: “...included [only] adults with...”
10. P4, L55: “imputed”; imputation has a fairly specific meaning, relating to missing or unknown data. I don't think this is what you mean.
11. P5L31: Please explain what 'sensitive cases' are. This was a sizeable exclusion and potentially highly relevant to ACE.
12. P5L37: What were the reasons for the 34% attrition between sweeps 1 and 7?
13. P5, L43: Why did you not use IPW or MI, given the strong associations between ACE and cohort attrition in other studies (e.g. <http://doi.org/10.14301/llcs.v8i4.414>)? Did you explore differences in characteristics of complete vs incomplete cases?
14. P5, L46: GUS acronym not defined
15. P5, L48: “linked” implies joining separate data sources, not separate waves from within one study.
16. P5, L55: terms like “ACEs 1 and 5” are neither transparent nor consistent with the literature. Suggest defining and consistently using terms physical abuse, sexual abuse, etc.
17. P5, L56: “ACE 3 (sexual abuse) was included in an open ended question”. Looking at the supplementary table, which explains that there was a question ('any other adverse events'), I do not think you can say that sexual abuse was “included”.
18. P6, L21: Why did you reduce ACE to a binary measure and why did you use these thresholds?
19. P6, L23: It is not clear what correlations you were examining or why. Findings (“all correlations were of a weak to medium strength”) should not be reported in this section and are not sufficiently explained to be understood anyway (i.e. what is “weak” or “medium”?)
20. P6, L28: Please make it clear when talking about income that this is at sweep 1/baseline/etc.
21. P9, L11: It is not possible for the reader to tell if the model for 3+ ACEs was similar or not because the authors have excluded all of the non-significant variables. Because 3+ ACEs is rarer than 1+ ACEs, the model is going to have less power so it is not unexpected that some of the predictors may become 'nonsignificant'. Significance is largely irrelevant here; it is the effect sizes that the authors should be more interested in.
22. P10, L17: What is a “complex samples module”?
23. P11, L36: This paper is certainly not “unique in using a range of prospective data on ACEs”; there are several large population-based cohorts around the world that have produced similar analyses.
24. P12, L54: “longitudinal weights”; was the analysis actually weighted? This is the first time it is mentioned. If so then by what characteristics?
25. P12, L67: “The first is recall”; the first reason for differences is almost certainly differences in the definitions of ACEs implied by the phrasing of questions and coding of responses; i.e. the studies are measuring different things.
26. P12, L63: “had high prevalence rates in this cohort, which may bias...” Odds ratios are not biased relative risks; they are different ways of expressing ratios of probabilities, which diverge at higher levels of probability.

	The authors could easily have converted OR into RR if they wished. 27. P12, L60: "...as well as the strength of associations between poverty and ACEs"; I agree with the statement about prevalences but it is very difficult to predict what effect selection bias will have on an association. It can easily go either way. 28. P17, Figure 1: Y axis not labelled. Title has poor grammar. No caption. Figure does not seem justified given the small amount of information that it illustrates. 29. P18, Figure 2: No caption, insufficient contrast for legibility. 30. P19, Supplementary table: No caption. Acronyms not defined. Source of ACE questions not cited.
--	--

VERSION 1 – AUTHOR RESPONSE

Reviewer: 1

Comments to the Author

I have conducted a statistical review of the manuscript "What factors are associated with Adverse Childhood Experiences in Scottish children: results from a birth cohort study."

The authors investigate which factors are associated with adverse childhood experiences using a prospective cohort of over 3000 children in Scotland. The authors present the first analysis to investigate ACEs using a prospective approach.

I am satisfied that the statistical approach taken by the authors is appropriate for the objectives and data in question.

I have made a few comments where I feel wording or terminology is unclear, or where I feel additional information would be helpful for a reader:

1) Introduction, page 4, line 53: "The aims of this paper are to explore to what extent ACEs could be imputed using prospective cohort data,"

I'm not sure that 'imputed' is quite the right word here, perhaps 'investigated' or 'predicted' (if the aim is to determine ACEs before they have happened).

Line 102 - We have changed this to 'determined'

2) Methods, page 5, line 12 "Data were taken from the first seven waves of the Growing Up in Scotland Study..."

Please define the GUS abbreviation here (it is used in the paragraph below).

Line 107 – this oversight has been corrected.

3) Methods, page 5. For context, can the authors please add a sentence or two on the aims/objectives etc. of the Growing Up in Scotland Study? I assume that the study was conducted for wider objectives than determination of ACEs?

Yes, you are correct that the study had a very wide remit to capture information on the lives of children growing up in Scotland. We have added a sentence in on this in lines 108-112, which we hope explains this further.

4) Methods, page 5, line 14: What is the sample clustered by?

The sample is clustered by datazone. We have reworked this section, and referenced the data user guide, where sampling is fully described, to try to make it clearer in the light of all reviewer comments [lines 112-124]

5) Methods, page 5, line 54: "Seven ACEs (or proxies for them) were available..." It would help for context to add out of how many possible ACES. This is implied in the rest of the paragraph but it would be helpful to make it explicitly clear:

i.e. "Seven ACES (or proxies for them) out of 10 possible ACE questions were available..."

Good point - this has been changed.

6) Analyses, page 6, line 21: "Binary variables of 1+ ACE (vs. none) and 3+ACEs (vs. <3) were derived."

Some of the notation here is a little confusing, on first reading I thought this was a calculation (i.e. 1 plus the number of ACE questions). To avoid misunderstanding throughout these methods and results, I suggest using words rather than + signs (i.e. one or more ACE vs no ACEs, at least 3 ACEs vs less than 3 ACEs etc.).

Lines 144-145 – we did this purely due to lack of space in terms of word count. We have now defined each one at the start of the analysis section, which we hope will be acceptable.

7) Analyses, page 6, line 21: "Data were described and Spearman correlations explored. All correlations were of a weak to medium strength."

Which data is being described here in terms of correlations? I presume that this data was skewed (i.e. non-normally distributed) due to the use of non-parametric Spearman rank correlation?

We have changed this to say 'socio-demographic data', which we hope is clearer. We have provided a table of the correlations. Data were either non-normally distributed or ordinal.

8) General comment, analysis and results. Please note that logistic regression produces odds ratios (rather than risk ratios) and therefore results should be interpreted in terms of odds rather than risk i.e. please reword 'predicting the risk' (page 6, line 26), "...associated with heightened risk" (page 9, line 3) and so on.

You are quite right – these have been corrected throughout.

9) Analysis, page 6 and 7: The terms used in the calculation for population attributable risk and the terms in the footnotes of this calculation are different (P and Pe, RR and RRe), please use consistent terminology.

We have changed this.

10) Results and Discussion. Minor point, but 65% is not two thirds. Please reword to 'about two thirds' or 'nearly two thirds' or just quote the proportion

Line 167 – this has been changed to 'Around two-thirds'.

11) Results: It would be helpful if the numbers reporting each of the seven ACES could be reported in a table / on a Figure. Most of the results refer to the number of ACEs reported rather than specifics. Some information is given in the second paragraph of the results regarding specific ACEs but complete details of this would be interesting to see

We have added this as a Supplementary table – see Supplementary Table 2

12) Table 2: Please also report numerical results (odds ratios and confidence intervals) for the non-significant results. The magnitude of effect (as well as statistical significance) may be of interest to readers.

These figures have been added.

13) Table 2: R-squared values (the amount of variability explained by the covariates in the regression model) are relatively low and a lot of variability remains. This is not a shortcoming of the authors or the methods, it is the reality of attempting to find statistical associations where relationships are complex and likely confounded. Do the authors have any thoughts on any factors (known or unknown, perhaps information that was not collected within the GUS) study which could also be influencing ACEs? Could the authors add a couple of sentences to the discussion regarding the remaining variability and, if known, any potential reasons for this?

Lines 282-284 – we are pretty short on space to expand too much on this but have added a sentence on our views regarding the remaining variability.

14) Figure 1 and Figure 2, Table 2 – I assume that 'base' refers to the sample size of the data displayed within the figure/table? Perhaps use a different term for clarity

This has been altered.

15) Figure 1: I assume this title should read "Number of ACEs experienced at age eight"?

Sorry – yes- this has been changed.

16) Figure 2: It is quite difficult to read the numbers (I assume proportions %?) on bars, particularly those close together towards the top. Perhaps these % could be displayed next to the bar rather on top, or in a table below the figure?

Reviewer 2 has pointed out that the data in Figure 2 is already in Table 1 so we have deleted this figure.

17) Results, page 11, line 3 "As poverty appeared to play such an important role in children's likelihood of experiencing ACEs..."

We have deleted this section in response to reviewer comments and lack of space.

18) Please add details into the background of the association between poverty and ACEs with any references add needed.

A brief overview has been added - unfortunately we are unable to expand further due to the word limit.

19) Results, page 11: Please also provide the numbers of the sample living in relative poverty with and without at least one ACE (so that a reader has enough information to perform the calculation in the methods, if desired).

This has been added in Lines 216-218.

Reviewer: 2

Comments to the Author

This study documents the prevalence of adverse childhood experiences (ACEs) using prospectively-collected data from the Growing Up in Scotland Study, and examines early childhood risk factors for ACE experiences.

This is a worthwhile contribution to the literature, but suffers from lack of clarity in some areas. These will need to be addressed before this paper can be considered for publication. Examples of this are:

1. In the final paragraph of the introduction it is stated, "The aims of the paper are to explore to what extent ACEs could be imputed using prospective cohort data, and what prevalence looks like in a recent population". No imputation occurred from what I could tell. Isn't the aim just to "determine the prevalence of ACEs using prospective cohort data"?

Line 102 - We have changed this to 'determined'

2. I found the description of the sample confusing. From the description, it seems there were PSUs of aggregated data zones, each with an average of 57 live births/year. How were births chosen from these PSUs, and what role did the stratification by local authority and imd score play? And how many births did this result in sampling, and what steps were there in sampling the 5,217 assessed at sweep 1? What was 'sensitive' about the sensitive cases that meant they had to be removed? No doubt the sample has been described many times in Growing Up in Scotland publications, so perhaps a baseline GUS study can be referred to, but the description in this paper still needs to be clearer, in my view.

We have reworked this section, and referenced the data user guide, where sampling is fully described, to try to make it clearer in the light of all reviewer comments [lines 112-124]

We have given an example of sensitive cases, which we hope will make this clearer.

3. At the end of the first paragraph of the methods it is stated, "Cases without full data were not used: no imputation was used." The obvious question arises whether the 10% not assessed differ from the 90% assessed. From the discussion and ref 17 it looks like there was differential loss to follow-up. How exactly did the 10% differ from the 90%?

We have added this information into lines 136-138.

4. It is not stated who reported on the ACEs. In the strengths section of the discussion it is indicated that "a mixture of parental and child-reported data" were used, but this isn't described anywhere. Who reported on which ACE?

We have added a column in Supplemental Table 2 with this information. We haven't included it in the body of the text due to space limitations.

5. In the analyses section it appears that variable selection took place, but Table 2 seems to include all predictors. What was the procedure? My personal preference is for no selection and to show univariable and multivariable associations, but I'll leave that to the authors.

We have removed the selection and added in non-significant figures, in light of all reviewer comments.

6. The PAR analysis at the end of the results comes across as a little ad-hoc, and the interpretation assumes the poverty-ACEs association is causal, which is moot. I'd be inclined to remove this unless there's a specific reason for inclusion that is addressed in the introduction.

We have removed this section.

7. The discussion mentions longitudinal weights, but the methods do not indicate how these are used (and calculated).

Apologies – this was an oversight on our part. Information on weights used has been added to the methods section – Lines 131-134.

There are also several minor issues that the authors may wish to address:

1. Under 'What the study adds', it is stated that, "At age 8, children had higher levels of one ACE". Higher than what? If 'high' is meant, perhaps state the actual proportion or percentage with 1+ and 4+.

We have re-written this sentence.

2. The 'GUS' acronym isn't described in the text. Maybe do so in the first line of the methods.

Line 107 – this has been described.

3. I don't understand the 'e' in the PAR formula.

We have deleted this section.

4. The education and deprivation associations in Table 1 aren't discussed in the text, where all other associations are. Why?

This was an omission which has been rectified in lines 195-197. We are unable to expand on this unfortunately due to lack of space.

Also, figure 2 repeats the income findings from Table 1. Why?

This is a good point! We have deleted this figure.

5. In Table 2, the r-squared and 'base' (is this the n?) are shown in the odds ratio column. Maybe these could be at the top of the table or footers.

We have changed 'base' to sample size to make this clearer. We have put an extra row and a strong black line, as well as italicising the n and r-sq numbers to differentiate this more clearly.

6. Under strengths, it's unclear why the English and Welsh ACE studies are specifically targeted when highlighting the GUS response rate as a strength. Perhaps instead compare the GUS response rate to a range of international studies.

This is a fair point. We highlighted these as they are some of the most recent and based in local context. We have added in the original Felitti response for comparison: response rates were surprisingly difficult to obtain from many papers.

7. In the conclusion, perhaps add "during their life" to the end of the first sentence.

This has been done.

Reviewer: 3

Comments to the Author

General comments:

This manuscript describes an analysis of risk factors for adverse childhood experiences (ACE) in a Scottish birth cohort, including estimation of the proportion of ACE attributable to poverty. As the authors indicate, there are few prospective studies in the ACE literature. There are also few on the effects of poverty on ACE. Unfortunately, though, the analysis is inadequately reported, poorly integrated with the existing literature, and suffers from several statistical limitations and errors. My major concerns include:

1. The description of the methods is unclear. Initial statements like “derived from child benefit records” (P5, L14) seem to imply that the study relied on administrative data. Then, later in the same paragraph there is a statement “the target interview date was 94.5 months old”. Was this for sweep 7? There is no explanation at all of data collection between sweeps 1 and 7. Were participants interviewed at every sweep, or was it some blended design of administrative and interview/survey data? Who was interviewed – the parents and/or children? (it is indicated later that both were interviewed but the details about when and how are not at all clear).

We have added in more detail on this in both the methods section and in Supplementary Table 1.

2. Statistical problems include that:

a. Missing data are ignored, despite known/expected associations with ACE. There is no explanation for what happened to 34% of the cohort between sweeps 1 and 7, no comparison of characteristics between those lost and those retained, or those incomplete (another 10%) and those complete. Really, there should be at least an attempt at inverse probability weighting or multiple imputation but even a descriptive analysis is missing.

We have added in a brief descriptive analysis of the missing data in the 10% incomplete cases (Lines 136-138). Data are weighted using a longitudinal weight and information has been added in about this at Lines 131-134. When using these data we have to weigh up whether to use multiple imputation, or to account for the complexity of the sample, as the package we use cannot do both. We chose to account for the stratification alongside using the longitudinal weights.

b. A simple, poorly described, automated variable selection procedure was implemented to build the model according to the significance of individual parameters. Automated variable selection is better suited to building predictive models from large numbers of available predictors, not for adjusting for confounding in causal effect estimation, which appears to be the aim of this analysis (specifically, estimating the effect of poverty on ACE risk). Estimating causal effects requires a conceptual understanding of confounding (i.e. justification for the selection of covariates, more than just that they significantly predict the outcome) and other sources of bias (e.g. missing data), with appropriate statistical methods for addressing them. Crucially, you should not drop confounders or explanatory variables just because they are not significant.

We appreciate that we have not explained this clearly enough and have tried to add in more detail within the word limits. We have left in the non-significant items in light of previous reviewers comments.

c. Attributable fractions have been calculated using an inappropriate formula; the formula for use when there is no confounding, rather than that for use with adjusted relative risks. See Rockhill B, Newman B, Weinberg C. Use and misuse of the population attributable fraction. *Am J Public Health*. 1998;88(1):15-9; or Gefeller O. Comparison of adjusted attributable risk estimators. *Stat Med*. 1992;11(16):2083-91.

In light of other comments we have decided to remove this section.

d. The interpretation of results indicates a poor understanding of the differences role of statistic significance testing vs effect sizes, the relationship between odds ratios and relative risks, the interpretation of attributable risk, the interpretation and use of pseudo-R2 (mistakenly referred to as R2) in logistic regressions, and the even the implied use of R2 when the aim of modelling is not prediction or construction of a full explanatory model.

Given these multiple limitations in the analysis and interpretation of results, I strongly suggest that the authors seek statistical support before revising.

We believe we have addressed these comments.

3. The 'mapping' of adverse childhood experiences from the GUS interview responses is poorly described and inconsistent with the literature. For example, the authors seem to have equated smacking "when [the child] has done something wrong" (often or always) with physical abuse. This is highly contentious and not consistent with other definitions used for physical abuse, including the one cited for comparison. The threshold level of >14/units per week that the authors have set for mapping parental alcohol problems, while associated with increased health risks, would not normally be considered indicative of problem drinking in the context of ACEs. Do the authors know what proportion of UK adults/parents drink at this level and how those statistics relate to those recorded in the study? And how these relate to parental alcohol problems recorded in other ACE studies?

We feel that, given the available space, we have been clear both about the items used to assess each ACE (Supplementary Table 1), and that the use of this sort of proxy information is a weakness of the study (Lines 235-236). We agree that this method is contentious, and that we have had to decide what information best aligns with the original ACE questionnaire, as well as where to implement cut-offs. Where there has been some sort of independently agreed cut-off (such as in the case of harmful drinking) we have used this cut-off; in other cases, such as in the Depression, Anxiety and Stress Scale, where there is no validated cut-off, we used standard deviations to devise cut-offs. The Physical abuse measure is one which was debated by the authors. The original scale leaves it up to the adult being asked retrospectively to determine whether "smacking" should be included – it is only latterly (and I believe in the UK) that this has been clarified in brackets not to include 'gentle smacking'. We have an issue with this for two reasons: firstly that this appears to be a moral judgement, rather than an objective one, based around the on-going belief in the UK that parents should be allowed to hit their children, as long as it doesn't leave a mark (whether this is immediately or in the long-term is usually undefined); secondly, we now have substantial evidence about the harm resulting from smacking, particularly in terms of mental health. Whilst we are not equating all smacking with abuse, the line is possibly more grey than it at first appears. Whilst we would like to more thoroughly discuss all these issues in the paper, with a 2,500 word limit, we are heavily restricted, and thus provide the information for readers to make up their own minds.

4. The prevalences of each ACE are not even reported, making it impossible to assess their consistency with other research or the contribution of each ACE to the number of ACEs.

We have added this as Supplementary Table 2.

5. The introduction is entirely missing references to the existing literature on risk factors for ACE and the impact of poverty on ACE.

We have added a brief (due to the word limit) section in on this existing literature.

The discussion is dismissively critical of previous research, to the point of seeming unjustified or unkind in places.

We feel that we have objectively compared and contrasted our research from other recent ACE studies. It would be helpful if you could highlight which particular paragraphs you feel are unjustified or unkind.

Other comments and questions:

6. P2L6 (what this study adds): 'higher' than what? Suggest rephrasing.

This has been rephrased.

7. P3L18 (Design): Logistic regression models were fitted to explore associations – adjusted for what?

We have added this information in (Line 67-68)

8. P4: Opening sentence grammar needs improving at “, and”, e.g. “, who were”.

This has been changed.

Also, acronyms should not have 's' appended in plural form (but check with journal guidelines).

We can't see any guidelines within the journal for this – perhaps the editor has a view. We think that the text reads better as 'ACEs' rather than 'ACE' however we are happy to change this if necessary.

9. P4, L27: “...included [only] adults with...”

Line 93 – we have added 'only' in.

10. P4, L55: “imputed”; imputation has a fairly specific meaning, relating to missing or unknown data. I don't think this is what you mean.

Line 105 – we have changed this to 'determined'

11. P5L31: Please explain what 'sensitive cases' are. This was a sizeable exclusion and potentially highly relevant to ACE.

Lines 124-126 – we have added some additional information in on this.

12. P5L37: What were the reasons for the 34% attrition between sweeps 1 and 7?

The reasons for attrition are not described within the data documentation. It is mentioned that some families move outwith Scotland, making them ineligible, but no other reasons are given.

13. P5, L43: Why did you not use IPW or MI, given the strong associations between ACE and cohort attrition in other studies (e.g. <http://doi.org/10.14301/Ilcs.v8i4.414>)? Did you explore differences in characteristics of complete vs incomplete cases?

Please see answer to 2a.

14. P5, L46: GUS acronym not defined

We have defined this on Lines 110-111.

15. P5, L48: “linked” implies joining separate data sources, not separate waves from within one study.

Line 140 – we have changed this to 'combined into one dataset'

16. P5, L55: terms like “ACEs 1 and 5” are neither transparent nor consistent with the literature. Suggest defining and consistently using terms physical abuse, sexual abuse, etc.

Lines 144 and 145 – this sentence has been re-phrased.

17. P5, L56: "ACE 3 (sexual abuse) was included in an open ended question". Looking at the supplementary table, which explains that there was a question ('any other adverse events'), I do not think you can say that sexual abuse was "included".

Line 145 – this sentence has been re-phrased.

18. P6, L21: Why did you reduce ACE to a binary measure and why did you use these thresholds?

We reduced the ACE score to a binary measure because we wanted these findings to be useful to policy-makers and practitioners in Scotland, and we believe that, from our experience, binary measures, along with odds ratios, are more meaningful for lay people (you may disagree). We used the 3+ cut-off was used instead of a 4+ cut-off due to having fewer items in the scale and less time to accumulate them. Using a 4+ cut off with this scale and age didn't provide enough power to examine factors associated with it adequately. 1+ was used because arguably the goal for policy-makers should be to prevent ACEs. We have added some additional information along these lines in Lines 155-157.

19. P6, L23: It is not clear what correlations you were examining or why. Findings ("all correlations were of a weak to medium strength") should not be reported in this section and are not sufficiently explained to be understood anyway (i.e. what is "weak" or "medium"?)

We have added the correlations in Supplementary Table, 2 along with a guide to their interpretation.

20. P6, L28: Please make it clear when talking about income that this is at sweep 1/baseline/etc.

This section has been deleted.

21. P9, L11: It is not possible for the reader to tell if the model for 3+ ACEs was similar or not because the authors have excluded all of the non-significant variables. Because 3+ ACEs is rarer than 1+ ACEs, the model is going to have less power so it is not unexpected that some of the predictors may become 'nonsignificant'. Significance is largely irrelevant here; it is the effect sizes that the authors should be more interested in.

We have added non-significant figures into the model.

22. P10, L17: What is a "complex samples module"?

This should have been removed prior to submission and has now been removed. The complex samples module is the part of SPSS which controls for attrition, clustering and weighting in the data.

23. P11, L36: This paper is certainly not "unique in using a range of prospective data on ACEs"; there are several large population-based cohorts around the world that have produced similar analyses.

Lines 226-227 – we have clarified this somewhat. We are aware of the Dunedin cohort and fragile families – are there others we have missed?

24. P12, L54: "longitudinal weights"; was the analysis actually weighted? This is the first time it is mentioned. If so then by what characteristics?

Apologies – this was an oversight on our part. Information on weights used has been added to the methods section – Lines 131-134.

25. P12, L67: “The first is recall”; the first reason for differences is almost certainly differences in the definitions of ACEs implied by the phrasing of questions and coding of responses; i.e. the studies are measuring different things.

You are correct that we mentioned this in study weaknesses but not in this part of the discussion. We have added a sentence in to this effect (lines 264-265). We weren't intending to rank order the reasons, however we have put this one first in case others also interpret this in the same way.

26. P12, L63: “had high prevalence rates in this cohort, which may bias...” Odds ratios are not biased relative risks; they are different ways of expressing ratios of probabilities, which diverge at higher levels of probability. The authors could easily have converted OR into RR if they wished.

We're not sure what you are wanting us to do here. We decided against changing ORs for RRs because we believe they are more challenging for lay people to interpret.

27. P12, L60: “...as well as the strength of associations between poverty and ACEs”; I agree with the statement about prevalences but it is very difficult to predict what effect selection bias will have on an association. It can easily go either way.

We have deleted this part of the sentence.

28. P17, Figure 1: Y axis not labelled. Title has poor grammar. No caption. Figure does not seem justified given the small amount of information that it illustrates.

We have removed the typo in the titled and labelled the axes. We have left in the figure as we believe this is useful for those out with academia to see this information visually.

29. P18, Figure 2: No caption, insufficient contrast for legibility.

This figure has been deleted as data already appears in previous table, as pointed out by reviewer 2.

30. P19, Supplementary table: No caption. Acronyms not defined. Source of ACE questions not cited.

This additional detail has been added.

VERSION 2 – REVIEW

REVIEWER	Reviewer name: James C. Doidge Institution and Country: University College London, UK Competing interests: None
REVIEW RETURNED	01-Oct-2018

GENERAL COMMENTS	My initial review of this manuscript focused on several methodological issues and a lack of integration with the existing literature. The revisions have addressed some of the methodological issues, for example by removing the section on economic determinants, but I think that there are deeper conceptual problems that underlie both of these concerns, which the authors would benefit from considering. It is still not at all clear what this study adds to the literature, and I think this is because of the treatment of 'adverse childhood experiences' as a distinct, tangible variable. 'Adverse childhood experiences' are a construct that was devised as a summary measure of developmental risk factors; different traumatic experiences that it was acknowledged have similar effects on developmental outcomes and flow-on effects into adulthood.
---

There are well-researched neurobiological and epigenetic mechanisms for explaining this similarity of effects. The same is not true when you treat ACE as an outcome; while some common factors such as poverty and intergenerational abuse may be implicated in most ACE, there are also very different sets of causes for things like parental separation and child sexual abuse, for example. Child factors are unlikely to have any role in the parent-level adversities (parental mental health, parental separation, incarceration, domestic violence and drug and alcohol problems) and can have associations in opposite directions among the different types of child maltreatment. This is highly relevant to the recording of ACE where, for example, this study recorded corporal punishment and emotional neglect (both more prevalence in boys) but not emotional abuse or sexual abuse (both more prevalent in girls). This is why there is very little literature on 'risk factors for ACE' (because it doesn't make much sense) but huge literatures on risk factors for different types of maltreatment and child maltreatment in general, for domestic violence, parental mental health problems, etc. By focusing on ACE as an outcome, this article ignores all of that relevant literature.

The big question is: what question is this study trying to answer? Is it a prevalence study, a risk prediction study, or a study of the causal mechanisms underlying childhood adversity? As it stands, it seems to be trying to cover all bases but does not do a very good job of any. As a prevalence study for overall ACE scores, there are too many differences in how ACE were recorded to be comparable with other studies. My suggestion is that, if you want to focus on prevalence estimation then you focus on those specific adversities for which measurement is most comparable. If you want to look at risk factors or causes of adversities, then it makes more sense to focus on specific adversities, or groups of more comparable adversities. If you want to examine risk factors for adversities, do you want to find the best predictors in this study, or interpret their role as causal determinants? This has implications for your analysis design, selection of variables, etc. Currently, it is not clear what the analysis is aiming achieve, or how the analysis design supports those objectives. This lack of rationale and a clear link from objectives to design, makes the study appear as just research for the sake of doing research. These problems are highlighted by the 'What is known about the subject' text, which describes outcomes of adverse childhood experiences, and the 'What this study adds' text, which lists some risk factors for ACE that are already well-established in the literature (and also differ across specific adversities). These sections are an opportunity for the authors to distil their core message, and they are currently unlinked from each other and contain nothing particularly new or interesting.

Aside from the conceptual issues, there remain outstanding statistical problems that were highlighted in the initial review: most importantly misinterpretation of nonsignificant p-values as implying an absence of association or effect, but also the more minor issues of misinterpretation of pseudo-R² as percent variance explained (and misrepresentation as R²), and misinterpretation of the implications of high prevalence for odds ratios as leading to bias. The issue of missing data has been addressed with only a brief textual explanation of an "exploratory analysis", with no statistics provided to indicate the extent of any possible problems here.

Prevalence estimates are particularly sensitive to selection bias so if the authors wish to present this paper as a prevalence study, then the issue of missing data should be addressed in more detail.

The authors expressed concern about the "haughty and condescending" tone of my initial review, fearing how it might have been received by an ECR. I wrote that review while feeling frustrated at the unnecessarily long time that is required to review a paper of that quality, at the disregard of the relevant literature on risk factors for child maltreatment and other adversities, and with knowledge that the authors were not ECRs. I hope that the authors can appreciate the time that is required (and volunteered) to provide a reviews as detailed as these, and that they use them to increase the value of the paper. It remains an important topic for research.

Specific comments:

1. P1: What is known about the subject: I don't understand why this focuses on outcomes of ACE when the study appears focus on prevalence and risk factors for ACE.

2. P2: What this study adds: These are well-established risk factors for ACE that could just as easily have been listed under "What is known about the subject". These sections are your opportunity to highlight the key contribution of your study.

3. P3: Abstract: It would be helpful if the abstract mentioned the Growing Up in Scotland study instead of 'Child Benefit Records', which is misleading (even if technically correct). It would also be helpful if it stated which adverse experiences were actually analysed, rather than that "three of the 10 couple [sic] not be assessed". There is no list of 10 standard ACEs. The number and definition of ACEs varies considerably across studies, even those aiming for consistency with the Felitti et al.

4. P4L87-88: "Whilst there is likely to be a substantial overlap with deprivation, this is generally unknown" There is a large body of literature on the topic of the relationship between deprivation or socioeconomic disadvantage, child maltreatment and other adverse childhood experiences. There are systematic reviews and books on the subject. I don't think that this manuscript is adequately integrated into this literature.

5. Opening sentence: "Adverse Childhood Experiences (ACEs) were first explored among US adults". While I appreciate that this study was seminal in bringing ACE into the public health space, there was wealth of other research on the topic over the fifty years leading up to that.

6. P5L114: "was derived from child benefit records". Should be something like "identified using"; 'derived from' implies that the study was based on data from them.

7. L141: "out of a potential 10" – suggest removing this for the reasons discussed above

8. L128-133: The exploratory analysis of missing data that has been added goes some way towards addressing this issue but appears to focus on the 10% of wave 7 respondents with incomplete data, leaving out the 34% who were lost between waves 1 and 7.

However no statistics are provided to give readers any sense of the extent of the problem and there is still no side-by-side comparison of characteristics.

9. L137-145: I think this text should clearly describe what was actually measured (i.e. not just what wasn't measured), and that supplementary tables 1 and 5 be included in the main document because they are critical to understanding this. Those tables could be merged for efficiency if it helps.

10. L160-162: This text describes a variable selection process still being used.

11. L170: Reference to Supplementary Table 3 should be 5.

12. I disagree with the interpretation that similar patterns were seen in the White ethnic group compared with other ethnic groups. This is a textbook misuse of 'non-significant' p-values as implying the absence of relationship, and one of the reasons that I suggested that the authors seek statistical support before revising. The observed association was quite strong (OR = 1.5 and 1.8) and in the opposite direction to what might be expected (higher risk in White children), which warrants checking and explanation.

13. L189: Noting a difference in the role of maternal education being 'nonsignificant' in the 3+ model is another misuse of p-values. This difference between models was not tested and the point estimates are quite similar. Treating maternal education as being multinomial was a poor choice too, when it is effectively ordinal.

14. L192: As I pointed out last time, there is no such thing as variance of a categorical outcome; this is misreporting and misinterpretation of pseudo-R² (i.e. you cannot interpret pseudo-R² as percent of variance explained). Supplementary Table 3 also includes this misreporting of pseudo-R² as R².

15. L206-208: The authors talk about the study as having a high response rate, presumably referring to the 90% of those not already lost to follow-up, which was another 34%. This doesn't seem particularly high to me, and doesn't account for those who didn't respond at baseline; it's not a fair comparison.

16. L230-231: As indicated last time, high prevalence rates do not "bias odds ratios away from the null"; at higher levels of prevalence, odds ratios diverge from relative risks but are not "biased" and this is not a "weakness" of the study. This is a misunderstanding of so-called 'rare disease assumption' and the relationship between risk ratios, odds ratios and null hypotheses.

17. L257-263: The authors talk about boys having higher risk in this study and how this is the opposite from some other studies. This study focused mostly on parental ACEs: 5 of the 7 had nothing to do with the parent-child relationship; only corporal punishment and emotional neglect did. These are the two types of child maltreatment known to be experienced more by boys, while girls report higher rates of sexual and emotional abuse, which were not able to be analysed. Thus, the findings are actually not inconsistent with the literature. This illustrates one of the problems with this approach of considering ACEs as a distinct outcome, rather than as a summary of related risk factors as it was originally intended.

VERSION 2 – AUTHOR RESPONSE

My initial review of this manuscript focused on several methodological issues and a lack of integration with the existing literature. The revisions have addressed some of the methodological issues, for example by removing the section on economic determinants, but I think that there are deeper conceptual problems that underlie both of these concerns, which the authors would benefit from considering. It is still not at all clear what this study adds to the literature, and I think this is because of the treatment of 'adverse childhood experiences' as a distinct, tangible variable. 'Adverse childhood experiences' are a construct that was devised as a summary measure of developmental risk factors; different traumatic experiences that it was acknowledged have similar effects on developmental outcomes and flow-on effects into adulthood. There are well-researched neurobiological and epigenetic mechanisms for explaining this similarity of effects. The same is not true when you treat ACE as an outcome; while some common factors such as poverty and intergenerational abuse may be implicated in most ACE, there are also very different sets of causes for things like parental separation and child sexual abuse, for example. Child factors are unlikely to have any role in the parent-level adversities (parental mental health, parental separation, incarceration, domestic violence and drug and alcohol problems) and can have associations in opposite directions among the different types of child maltreatment. This is highly relevant to the recording of ACE where, for example, this study recorded corporal punishment and emotional neglect (both more prevalence in boys) but not emotional abuse or sexual abuse (both more prevalent in girls). This is why there is very little literature on 'risk factors for ACE' (because it doesn't make much sense) but huge literatures on risk factors for different types of maltreatment and child maltreatment in general, for domestic violence, parental mental health problems, etc. By focusing on ACE as an outcome, this article ignores all of that relevant literature.

The big question is: what question is this study trying to answer? Is it a prevalence study, a risk prediction study, or a study of the causal mechanisms underlying childhood adversity? As it stands, it seems to be trying to cover all bases but does not do a very good job of any. As a prevalence study for overall ACE scores, there are too many differences in how ACE were recorded to be comparable with other studies. My suggestion is that, if you want to focus on prevalence estimation then you focus on those specific adversities for which measurement is most comparable. If you want to look at risk factors or causes of adversities, then it makes more sense to focus on specific adversities, or groups of more comparable adversities. If you want to examine risk factors for adversities, do you want to find the best predictors in this study, or interpret their role as causal determinants? This has implications for your analysis design, selection of variables, etc. Currently, it is not clear what the analysis is aiming achieve, or how the analysis design supports those objectives. This lack of rationale and a clear link from objectives to design, makes the study appear as just research for the sake of doing research. These problems are highlighted by the 'What is known about the subject' text, which describes outcomes of adverse childhood experiences, and the 'What this study adds' text, which lists some risk factors for ACE that are already well-established in the literature (and also differ across specific adversities). These sections are an opportunity for the authors to distil their core message, and they are currently unlinked from each other and contain nothing particularly new or interesting.

Aside from the conceptual issues, there remain outstanding statistical problems that were highlighted in the initial review: most importantly misinterpretation of nonsignificant p-values as implying an absence of association or effect, but also the more minor issues of misinterpretation of pseudo-R² as percent variance explained (and misrepresentation as R²), and misinterpretation of the implications of high prevalence for odds ratios as leading to bias. The issue of missing data has been addressed with only a brief textual explanation of an "exploratory analysis", with no statistics provided to indicate the extent of any possible problems here.

Prevalence estimates are particularly sensitive to selection bias so if the authors wish to present this paper as a prevalence study, then the issue of missing data should be addressed in more detail.

Response: We believe that we have addressed the comments listed in this paragraph specifically in the points below: we have deleted some items of disagreement and, where we felt others had been misinterpreted due to poor wording on our part, we have rewritten sections; we have also added a new supplementary table detailing the demographics of sweep 7 missing cases. We hope this will allay any concerns.

Comment: The authors expressed concern about the "haughty and condescending" tone of my initial review, fearing how it might have been received by an ECR. I wrote that review while feeling frustrated at the unnecessarily long time that is required to review a paper of that quality, at the disregard of the relevant literature on risk factors for child maltreatment and other adversities, and with knowledge that the authors were not ECRs. I hope that the authors can appreciate the time that is required (and volunteered) to provide a reviews as detailed as these, and that they use them to increase the value of the paper. It remains an important topic for research.

Response: As reviewers ourselves, we do indeed appreciate the time given to review peer's papers thoroughly and feel that the paper has been much improved thanks to all the reviewers' comments. We do not necessarily feel that we disregarded the literature, rather that the necessity to be brief, due to the very tight word limit given, alongside the complex nature of survey and measurements, and thus a longer methods section being required, meant that we were unable to provide a more nuanced discussion of the literature, as much as we would have liked to. Having witnessed junior colleagues on the receiving end of reviews with a similar tone, we felt it was our duty to express our concern at this practice, which we feel is, ultimately, unnecessary.

Specific comments:

1. P1: What is known about the subject: I don't understand why this focuses on outcomes of ACE when the study appears focus on prevalence and risk factors for ACE.

This is a fair point. We have altered the second sentence to focus on current evidence around risks.

2. P2: What this study adds: These are well-established risk factors for ACE that could just as easily have been listed under "What is known about the subject". These sections are your opportunity to highlight the key contribution of your study.

With the exception of younger mothers, we respectfully disagree with this. The previous evidence relates to deprivation in current circumstances (i.e. as an adult), whilst the urban association and male association (which we will discuss later in these comments) are new contributions, we believe. We have clarified this somewhat, by adding that we are referring to a current child cohort.

3. P3: Abstract: It would be helpful if the abstract mentioned the Growing Up in Scotland study instead of 'Child Benefit Records', which is misleading (even if technically correct). It would also be helpful if it stated which adverse experiences were actually analysed, rather than that "three of the 10 couple [sic] not be assessed". There is no list of 10 standard ACEs. The number and definition of ACEs varies considerably across studies, even those aiming for consistency with the Felitti et al.

We have added mention of the GUS study. With regards to the ACEs assessed, we have added more information into the results section on which ACEs were reported, and removed 'out of the ten potential' from the conclusions.

4. P4L87-88: "Whilst there is likely to be a substantial overlap with deprivation, this is generally unknown" There is a large body of literature on the topic of the relationship between deprivation or socioeconomic disadvantage, child maltreatment and other adverse childhood experiences. There are systematic reviews and books on the subject. I don't think that this manuscript is adequately integrated into this literature.

We appreciate what you say here. We are incredibly tight on space (the word limit in this journal being 2,500 words) and are struggling to add more nuanced discussion here without removing other sections, many of which were added in in light of reviewer comments at the previous review. We have added the following end to the sentence to indicate that we are aware of this: 'although evidence of associations with individual measures does exist'.

5. Opening sentence: "Adverse Childhood Experiences (ACEs) were first explored among US adults". While I appreciate that this study was seminal in bringing ACE into the public health space, there was wealth of other research on the topic over the fifty years leading up to that.

We have adjusted this to clarify that we are talking about the ACE scale, rather than individual items.

6. P5L114: "was derived from child benefit records". Should be something like "identified using"; 'derived from' implies that the study was based on data from them.

We have changed this.

7. L141: "out of a potential 10" – suggest removing this for the reasons discussed above

We have change this.

8. L128-133: The exploratory analysis of missing data that has been added goes some way towards addressing this issue but appears to focus on the 10% of wave 7 respondents with incomplete data, leaving out the 34% who were lost between waves 1 and 7. However no statistics are provided to give readers any sense of the extent of the problem and there is still no side-by-side comparison of characteristics.

We have added a table (the new Supplementary Table 1) in to give an overview of the demographics of missing cases at sweep 7 and referred to this in the text.

9. L137-145: I think this text should clearly describe what was actually measured (i.e. not just what wasn't measured), and that supplementary tables 1 and 5 be included in the main document because they are critical to understanding this. Those tables could be merged for efficiency if it helps.

The exact ACEs measured have been added to the text. We are not opposed to having (what is now) Supplementary Table 2 within the main body of the text, but would welcome the journal's view on this, as it is rather a large table? We have moved Supplementary Table 5 into the body of the text (now Table 1).

10. L160-162: This text describes a variable selection process still being used.

Thank you - this should have been removed.

11. L170: Reference to Supplementary Table 3 should be 5.

This is now Table 1 as it has been incorporated into the body of the text.

12. I disagree with the interpretation that similar patterns were seen in the White ethnic group compared with other ethnic groups.

This is a textbook misuse of 'non-significant' p-values as implying the absence of relationship, and one of the reasons that I suggested that the authors seek statistical support before revising. The observed association was quite strong (OR = 1.5 and 1.8) and in the opposite direction to what might be expected (higher risk in White children), which warrants checking and explanation.

We have not been clear here – we meant that similar patterns could be seen compared with the demographics listed in the previous sentence, not between the ethnic groups – however we can see how readers may misinterpret this and have therefore reworded the sentence. We suspect the rest of the comments were based on this misunderstanding, however to clarify, the p value did not meet the inclusion criteria set for the multivariable model, and we do point out that the power is limited by the small number of non-white children in the sample. We do not use the p-value to interpret the findings in any other way.

13. L189: Noting a difference in the role of maternal education being 'nonsignificant' in the 3+ model is another misuse of p-values. This difference between models was not tested and the point estimates are quite similar. Treating maternal education as being multinomial was a poor choice too, when it is effectively ordinal.

This was a poor choice of words and we have re-written this section. We do however feel that maternal education needs to be treated as a multinomial variable, as both the 'vocational' and 'other' categories mean that the variable is not strictly ordinal (e.g. some vocational qualifications, such as accountancy qualifications may be considered higher than a degree, whilst others may be considered lower).

14. L192: As I pointed out last time, there is no such thing as variance of a categorical outcome; this is misreporting and misinterpretation of pseudo-R² (i.e. you cannot interpret pseudo-R² as percent of variance explained). Supplementary Table 3 also includes this misreporting of pseudo-R² as R².

We have removed references to R²/pseudo-R².

15. L206-208: The authors talk about the study as having a high response rate, presumably referring to the 90% of those not already lost to follow-up, which was another 34%. This doesn't seem particularly high to me, and doesn't account for those who didn't respond at baseline; it's not a fair comparison.

That is a fair point and we have removed this sentence.

16. L230-231: As indicated last time, high prevalence rates do not "bias odds ratios away from the null"; at higher levels of prevalence, odds ratios diverge from relative risks but are not "biased" and this is not a "weakness" of the study. This is a misunderstanding of so-called 'rare disease assumption' and the relationship between risk ratios, odds ratios and null hypotheses.

We have removed this sentence.

17. L257-263: The authors talk about boys having higher risk in this study and how this is the opposite from some other studies. This study focused mostly on parental ACEs: 5 of the 7 had nothing to do with the parent-child relationship; only corporal punishment and emotional neglect did. These are the two types of child maltreatment known to be experienced more by boys, while girls report higher rates of sexual and emotional abuse, which were not able to be analysed. Thus, the findings are actually not inconsistent with the literature. This illustrates one of the problems with this approach of considering ACEs as a distinct outcome, rather than as a summary of related risk factors as it was originally intended.

We feel that this is a valid point and have added a sentence to this effect in the discussion.